# Rethinking Symbolic Regression: Morphology and Adaptability for Evolutionary Algorithms

**Kei Sen Fong** [1]**, Shelvia Wongso** [1]**, and Mehul Motani** [1,2]

[1] Department of Electrical and Computer Engineering, National University of Singapore

[2] N.1 Institute for Health, Institute for Digital Medicine (WisDM), Institute of Data Science, National University of Singapore

{fongkeisen,shelvia.w}@u.nus.edu, motani@nus.edu.sg

## Abstract

Symbolic Regression (SR) is the well-studied problem of finding closed-form analytical expressions that describe the relationship between variables in a measurement dataset. In this paper, we rethink SR from two perspectives: morphology and adaptability. *Morphology:* Current SR algorithms typically use several man-made heuristics to influence the morphology (or structure) of the expressions in the search space. These man-made heuristics may introduce unintentional bias and data leakage, especially with the relatively few equation-recovery benchmark problems available for evaluating SR approaches. To address this, we formulate a novel minimalistic approach, based on constructing a depth-aware mathematical language model trained on terminal walks of expression trees, as a replacement to these heuristics. *Adaptability:* Current SR algorithms tend to select expressions based on only a single fitness function (e.g., MSE on the training set). We promote the use of an adaptability framework in evolutionary SR which uses fitness functions that alternate across generations. This leads to robust expressions that perform well on the training set and are close to the true functional form. We demonstrate this by alternating fitness functions that quantify faithfulness to values (via MSE) and empirical derivatives (via a novel theoretically justified fitness metric coined MSEDI). *Proof-of-concept:* We combine these ideas into a minimalistic evolutionary SR algorithm that outperforms a suite of benchmark and state of-the-art SR algorithms in problems *with unknown constants added*, which we claim are more reflective of SR performance for real-world applications. Our claim is then strengthened by reproducing the superior performance on real-world regression datasets from SRBench. For researchers interested in equation-recovery problems, we also propose a set of conventions that can be used to promote fairness in comparison across SR methods and to reduce unintentional bias.

## 1 Introduction

Important discoveries rarely come in the form of large black-box models; they often appear as simple, elegant, and concise expressions. The field of applying machine learning to generate such mathematical expressions is known as Symbolic Regression (SR). The expressions obtained from SR come in a compact and human-readable form that has fewer parameters than black-box models. These expressions allow for useful scientific insights by mere inspection. This property has led SR to be gradually recognized as a first-class algorithm in various scientific fields, including Physics (Udrescu & Tegmark, 2020), Material Sciences (Wang et al., 2019; Sun et al., 2019) and Knowledge Engineering (Martinez-Gil & Chaves-Gonzalez, 2020) in recent years. The most common technique used in SR is genetic programming (GP) (Koza, 1992). GP generates populations of candidate expressions and evolves the best expressions (selected via fitness function) across generations through evolutionary operations such as selection, crossover, and mutation. In this paper, we rethink GP-SR from two evolutionary-inspired perspectives: morphology and adaptability.

**Morphology.** SR algorithms have traditionally used several man-made heuristics to influence the morphology of expressions. One method is to introduce rules (Worm & Chiu, 2013), constraints (Petersen et al., 2019; Bladek & Krawiec, 2019) and rule-based simplifications (Zhang et al., 2006), with the objective of removing redundant operations and author-defined senseless expression. An example is disallowing nested trigonometric functions (e.g. $sin(1 + cos(x))$). Another method is to assign complexity scores to each elementary operations (Loftis et al., 2020; Korns, 2013), intending to suppress the appearance of rare operations that are given a high author-assigned complexity score. However, these man-made heuristics may introduce unintentional bias and data leakage, exacerbated by the small quantity of benchmark problems in SR (Orzechowski et al., 2018). With the success of deep learning and its applications to SR, there exists substantial motivation to utilize deep learning to generate potential morphologies of candidate expressions in SR (Petersen et al., 2019; Mundhenk et al., 2021). Such a technique also comes with the benefit of being easily transferable to a problem with a different set of elementary operations. In this regard, we first show how current SR methods are reliant on these man-made heuristics and highlight the potential drawbacks. Then, in this paper, we show how using our neural network pre-trained on Physics equations (which we later introduce as TW-MLM) improves the performance of GP-SR even in the absence of such man-made heuristics.

**Adaptability.** SR algorithms tend to evaluate a candidate expression based on its faithfulness to empirical values. Some common fitness functions are Mean Absolute Error (MAE), Mean Squared Error (MSE) and Normalized Root Mean Square Error (NRMSE) (Mundhenk et al., 2021), among other measurements. We propose that SR should create a variety of characterization measures beyond the existing ones that measure faithfulness to empirical values. While previous work have suggested alternative characteristics in dealing with time-series data (Schmidt & Lipson, 2010; 2009), it is not easily transferable to SR in general and was not theoretically derived. In this paper, we propose to quantify the faithfulness of a candidate expression's empirical derivative to the ground truth. Motivated by evolutionary-theory (Bateson, 2017), we adopt an adaptability framework that changes the fitness functions across generations. This process makes it harder for pseudo-equations to survive through the different fitness functions and easier for a ground truth equation to survive across the generations. The additional benefit of such a method lies in the increased utility of the eventual expression. If the intention of the user is to take the derivative of the eventual equation, then a measure of faithfulness to empirical derivatives as fitness would assist that objective. In this paper, we alternate between different fitness functions to improve performance of GP-SR. Specifically, we alternate between MSE and a newly defined fitness function we term MSEDI.

**Proof-of-concept.** We combine these ideas to demonstrate a proof-of-concept (foreshadowed in Figure 1) through a minimalistic evolutionary SR algorithm that outperforms all methods (including state-of-the-art SR) in problems *with unknown constants* (i.e., *Jin\** (Jin et al., 2019)) and outperforms many benchmark models in problems *without unknown constants* (i.e., *Nguyen\** (Uy et al., 2011) and *R\** (Krawiec & Pawlak, 2013)). We contend that performance on datasets with unknown constants are more indicative of SR performance in real-world applications that including naturally occurring processes such as scaling. Our claim is then strengthened by reproducing this superior performance on real-world regression datasets from SRBench (La Cava et al., 2021).

To accommodate future research in SR using real-life datasets, we propose extra SR conventions for synthetic datasets in line with the "relevancy" criteria for results on benchmarks problem to correlate well with results on real-world applications (McDermott et al., 2012).

The remainder of this paper is organized as follows: Section 2 explains related work and mechanisms to improve SR, focusing on recent deep learning work. Section 3 describes our proposed methodology and mechanisms. Section 4 combines all our proposed mechanisms with the vanilla GP approach and compares the performance with state-of-the-art, commercial and traditional SR approaches. Reflections and future work are given in Section 5.

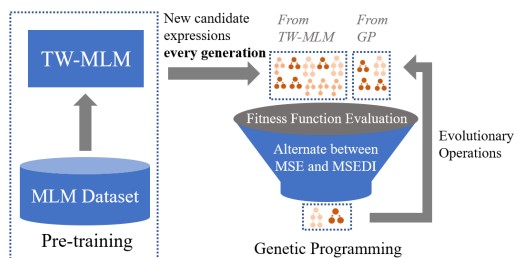

Figure 1: Integrated proof-of-concept schematic.

The main contributions of this paper are as follows:

1. We propose a set of *conventions and best practices* that reduce the risk of human bias in the context of using SR on real-life datasets.
2. We develop a predictive model called TW-MLM that learns the *morphology of expressions* through a mathematical language model, which is then used to generate candidate expressions for SR. TW-MLM serves as an alternative to man-made heuristics that is less prone to human bias and easily transferable to new problems.
3. We propose a method for alternating fitness functions inspired by *adaptability in evolution* theory to tackle the problem of multi-objective optimization in SR. We do this by alternating between MSE and our novel theoretically justified metric called MSEDI.
4. We develop an *integrated proof-of-concept* by combining our ideas and performed extensive testing on both synthetic and real-world datasets (from SRBench).

## 2 RELATED WORK

**Deep Learning for SR.** Deep learning has seen much success in SR, including Deep Symbolic Regression (DSR) (Petersen et al., 2019), which uses reinforcement learning to train a recurrent neural network (RNN) to complete an expression tree. Deep Symbolic Optimization Neural-Guided Genetic Programming (DSO-NGGP) is an improved method that combines a similar RNN trained by reinforcement learning coupled with GP (Mundhenk et al., 2021). The authors suggest that GP helps to produce large variations in the population of candidate equations when stuck at a local optimum. However, these methods are used in conjunction with several man-made heuristics to be effective. Other works include AI-Feynman, which utilizes neural networks to discover simplifying properties such as symmetry and separability to reduce the number of variables (Udrescu & Tegmark, 2020).

**Mathematical Language Model.** Symbolic mathematics has been successfully addressed as both a machine translation problem and a next word prediction problem (Lample & Charton, 2019; Kim et al., 2021). In particular, it has been reported that by adding a pre-trained seq2seq model as a supplementary mechanism to neural-guided SR, performance has improved (Kim et al., 2021). The consensus is that mathematics should be viewed as a language with complex rules. Most useful mathematical expressions are not only short and concise, they also tend to obey a variety of hidden rules, such as avoiding nested trigonometric functions (e.g. $sin(1 + cos(x))$. In this context, a mathematical language model can be developed to learn such rules implicitly.

**Derivatives in SR.** The usage of derivatives in SR has been hinted in previous works for time-series data (Schmidt & Lipson, 2010; 2009). In their work, given $N$ data samples from two time-dependent variables, $x(t)$ and $y(t)$, the system's derivatives are $\Delta x/\Delta y = x'/y'$, where $x'$ and $y'$ represent the empirical derivatives of $x$ and $y$ with respect to time. These values are then compared against the derivative obtained from the candidate expressions, $\delta x_i/\delta y_i$, through a mean logarithmic error: $-\frac{1}{N}\sum_{i=1}^{N}\log(1 + |\Delta x_i/\Delta y_i - \delta x_i/\delta y_i|)$. However, the theoretical derivation is not explored in their paper. Recent works in SR also include model discrimination by incorporating prior information on the sign of the first derivative based on physical knowledge (Engle & Sahinidis, 2021). In our paper, we develop a theoretical basis in line with existing assumptions for modelling errors to quantify faithfulness to derivatives for general equations which do not necessitate time-dependency.

**Multi-Objective Genetic Programming (MOGP).** Several general approaches exist for MOGP (Konak et al., 2006). A possible approach is to set all but one objective as constraints. However, the process of selecting values for constraints and which objectives to set as constraints is arbitrary. Another approach is to output a Pareto optimal set instead of a single individual expression to reflect the trade-offs between the different objectives. However, in the context of SR, the ground truth expression would be the best performer for both objectives, rather than forming a trade-off.

**Benchmarks Methods.** For comparison, we include traditional SR, state-of-the-art (SOTA), commercial algorithms, and random search. The methods selected report the lifetime population size of expressions to enable fair comparison of *recovery rates*. The benchmark methods are: (i) DSR (Petersen et al., 2019): Previous SOTA method that pioneered usage of reinforcement learning in SR; (ii) DSO-NGGP (Mundhenk et al., 2021): Current SOTA method in SR that is a hybrid of DSR and GP; (iii) GPLearn (Stephens, 2016): Python framework for standard GP-SR (Koza, 1992), which has seen wide usage as a generic SR method (Pankratius et al., 2018; Ferreira et al., 2019); (iv) TuringBot (TuringBot, 2020): Commercial SR product based on simulated annealing, which has been shown to be competitive among top commercial SR algorithms (Ashok et al., 2021). (v) Random Search: Generate expressions at random without evolution.

## 3 METHODOLOGY AND MECHANISMS

Here, we first propose revised conventions to be used for SR experiments targeted at resolving the flaws and criticism of current SR metrics, which are independent of our SR algorithm. We then introduce our novel methods of controlling morphology of expressions and promoting adaptability in evolution, and justify these methods individually through preliminary results and ablation studies.

### 3.1 PROPOSED CONVENTIONS AND BEST PRACTICES

We propose and justify conventions for SR experiments, adhering to the criteria discussed in the call for better benchmarking to be done so that SR experiments can correlate better with results on real-world applications (McDermott et al., 2012). In many recent papers, *recovery rate* has been utilized as the primary metric for evaluating SR and is defined as "the fraction of independent training runs in which an algorithm discovers an expression that is symbolically equivalent to the ground truth expression within a maximum of 2 million candidate expressions" (Mundhenk et al., 2021; Petersen et al., 2019; Kim et al., 2021; Larma et al., 2021). To this end, the conventions we propose will be focused on improving the effectiveness of *recovery rate* as an evaluation metric for SR experiments.

**Fixed set of primitive functions.** In contrast to previous SR experiments that use a varying primitive function set depending on the equation, we propose to use a fixed set of primitive functions across all datasets and all methods. This is a necessary step towards using SR on real-life datasets since we are blind to the underlying primitive functions in real-world scenarios. In our paper, we use a fixed primitive function set $\{add, mul, sub, div, sin, cos, arcsin, log, exp, pow\}$ for all datasets and methods, selected from the dataset used to train our mathematical language model.

**Top-1 Approximate Recovery.** We also propose a new measure, *top-1 approximate recovery rate*, that is more reflective of performance on real-life datasets compared to the exact *recovery rate* defined in the first paragraph of Section 3.1. *Top-1* means taking only the best scoring expression in the strictest sense, as consistent with how SR is used for real-life data (Abdellaoui & Mehrkanoon, 2021; Phukoetphim et al., 2016; Barmpalexis et al., 2011). In other words, we only assess one best equation per experimental run. We define an *approximate recovery* to be when the r-squared value of an expression (touted as the best error measure for SR (Keijzer, 2004)) over the entire sampling domain of the dataset is more than 99%. This is consistent with calls from the GP community to discourage measuring exact recovery on synthetic datasets as they usually do not correlate well with performance on real-life applications (McDermott et al., 2012). The most obvious drawback of an exact *recovery rate* is that for real-life datasets, it is impossible to measure the true recovery rate by checking mathematical equivalence since there will not be any accompanying ground truth equation for comparison. In our paper, we present *top-1 approximate recovery rate* as our primary metric, but we also include the results for *top-1 exact recovery rate* in brackets for comparison.

**Selecting appropriate lifetime population size.** To guard against setting a lifetime population that is too high, we also propose to benchmark against a random method (generate the entire lifetime population in one generation without any genetic operation) and to ensure the metric across all methods is not saturated at the selected population size. This is done to elicit meaningful conclusions from the results. Previous works have used an arbitrary lifetime population size of 2 million (Mundhenk et al., 2021; Petersen et al., 2019; Kim et al., 2021; Larma et al., 2021), but it is difficult to compare results across the different methods since many equations have near 100% recovery, even for the worst-performing methods. Additionally, using an arbitrary value may create unintended bias in results (Bergstra & Bengio, 2012). To these ends, we reduce the lifetime population size to 10000 following 2 observations we made prior to our main experiments. First, at the new size, the *top-1 approximate recovery rate* is never saturated across all methods, allowing us to compare the performance of each method across various equations. Second, we evaluate a fully random search, which is implemented in practice by setting GP with only 1 generation at full population size, and find that the performance across most of the equations is near 0. This gives us higher confidence to report that SR methods with positive performance do recover equations by pure chance.

**Datasets *with unknown constants* are more relevant.** We also recommend testing on datasets which include unknown constants, such as *Jin\** dataset, since we find that the performance of methods varies drastically with and without unknown constants. In addition, it is of practical interest to consider datasets *with unknown constants* since it is common in real-life relations between variables, such as feature scaling (Udrescu & Tegmark, 2020). For example, we have $f = e^{-\theta^2/2}/\sqrt{2\pi}$ a real-world physics equation from AI-Feynman database (see Appendix Table 7).

Table 1: Top-1 Approximate / Top-1 Exact Recovery Rates (%) of current methods across *Jin\** (with unknown constants) and *Nguyen\** datasets (without unknown constants). Results were averaged over 100 runs per equation per method.

| | DSO-NGGP | GPLearn | TuringBot | | DSO-NGGP | GPLearn | TuringBot |
|---|---|---|---|---|---|---|---|
| Jin*-1 | **47/0** | 29/0 | 1/0 | Nguyen*-1 | **95/75** | 3/1 | 59/5 |
| Jin*-2 | 2/0 | **23/0** | 1/0 | Nguyen*-2 | **74/43** | 15/6 | 34/0 |
| Jin*-3 | 12/0 | **16/0** | 0/0 | Nguyen*-3 | **81/12** | 15/2 | 28/0 |
| Jin*-4 | 0/0 | **21/0** | 0/0 | Nguyen*-4 | **81/5** | 15/1 | 19/0 |
| Jin*-5 | 7/5 | **14/3** | 1/0 | Nguyen*-5 | **10/3** | 0/0 | 8/0 |
| Jin*-6 | 0/0 | 0/0 | 0/0 | Nguyen*-6 | **88/44** | 1/0 | 39/0 |
| | | | | Nguyen*-7 | **99/2** | 57/0 | 98/0 |
| | | | | Nguyen*-8 | 18/3 | **46/6** | 39/10 |
| Jin*-Average | 11.33/0.83 | **17.17/0.50** | 0.50/0.00 | Nguyen*-Average | 68.25/23.38 | 19.00/2.00 | 40.50/1.88 |

In our experimental results recorded in Table 1, we observe that between TuringBot and GPLearn, TuringBot has the poorer performance in *Jin\** dataset that has unknown constants. However, TuringBot has the superior performance in *Nguyen\** dataset that does not contain unknown constants. A similar observation can be made between GPLearn and DSO-NGGP where the relative performance is reversed depending on the presence of unknown constants. This phenomenon is due to the difference in the frequency of appearance of constants and the presence and extent of constants optimization in each method. For datasets with no unknown constants, the more the algorithm utilizes and optimizes constants, the more likely that equations with morphology that are dissimilar to the true equation can become top candidates. This can be seen as inhibiting the evolutionary process, where the spots for expressions to undergo evolution are instead taken up by these psuedo-equations (expressions that perform well on the training set in terms of MSE but are not close to the ground truth equation in its functional form).

In the context of real-life datasets, we argue that it is justifiable to assume that unknown constants will appear frequently in naturally occurring processes such as scaling. The performance of methods on datasets without unknown constants would then be less reflective of the performance on real-life datasets. We thus recommend that SR experiments favor datasets with unknown constants and assert that results obtained from such datasets are more reflective of real-life application.

## 3.2 Morphology of Expressions

Previous works have been done to influence the morphology of candidate expressions using deep learning methods (Petersen et al., 2019; Bladek & Krawiec, 2019; Udrescu & Tegmark, 2020). However, these methods have never been fully independent, and have instead been an addition to another method or utilize man-made heuristics such as those mentioned in earlier sections. We also find that these methods can be heavily reliant on the heuristics. For instance, DSR performance drops sharply with the removal of in-situ constraints and complexity scores, with *top-1 approximate recovery rate* decreasing to 33% the original value and *top-1 exact recovery rate* decreasing to 10% the original value. Furthermore, these heuristics increase the likelihood for the algorithm to be biased towards certain form of expressions, allowing for an implicit data leakage. Finally, it is an extremely difficult task to form such man-made rules, and it is made even harder when the discovery of such rules needs to be repeated from scratch when the primitive function space is distinctively different, such as when changing from ordinary algebra to boolean algebra. Thus, we aim to propose a method that is free of such man-made heuristics. Here, we outline a standalone method of generating candidate expressions that can be used independently.

**Terminal walks representation.** Instead of using the prefix representation of expressions as input to a seq2seq model as done in other SR methods, we take inspiration from random walks used in node2vec (Grover & Leskovec, 2016) and generate terminal walks from expression trees to reflect the hierarchical nature of expressions. A single terminal walk refers to the collection of nodes traversed from the root of the expression tree to either a variable node or a constant node. These terminal walks will then be treated as sentences. The benefit of such a method over prefix representation is that the distances between operations in terminal walks are reflective of the distances between operations in expression tree. On the other hand, in prefix representation, the operations that appear to be faraway may instead be near in the expression tree. For example, for the equation $sin(1 + cos(x))$, the nested $cos$ is 2 tokens away in both the terminal walk $\{sin, add, cos, x\}$ and in the expression tree form. However, it is 3 tokens away in the prefix notation.

Table 2: Top-1 Approximate / Top-1 Exact Recovery Rates (in percentage) of GP and TW-MLM-GP across *Nguyen\**, *R\** and *Jin\** dataset. Results were averaged over 100 runs per equation per method.

|  | TW-MLM-GP (RNN) | TW-MLM-GP (TRANSFORMER) | GP |
|---|---|---|---|
| NGUYEN* | **24.13/7.25** | 23.51/7.31 | 19.00/2.00 |
| R* | **19.33/0.00** | 18.67/0.00 | 9.67/0.00 |
| JIN* | **21.33/0.50** | 17.67/0.50 | 17.17/0.50 |

**Building language model to replace heuristics.** We then treat the collection of all terminal walks as a corpus of sentences to train a RNN as commonly practiced in next-word-prediction natural language processing tasks (Barman & Boruah, 2018). For our paper, we used an embedding layer, a long short-term memory layer and a dense layer sequentially to train a lightweight RNN. This RNN will be our mathematical language model for our GP algorithm, which we coin as terminal walks mathematical language model (TW-MLM). Candidate expressions are then generated by randomly selecting an operation from a uniform distribution, then for every incomplete link in the tree, an incomplete terminal walk is generated and fed into the TW-MLM. The TW-MLM then outputs a *probability distribution* that is used to select the next node to complete the tree. This process repeats until the tree has no incomplete links.

**Ablation: TW-MLM to improve recovery rates.** In this ablation experiment, we use the baseline GP algorithm (Koza, 1992) implemented in GPLearn for comparison since the other competitive SR methods impose man-made heuristics which would complicate the insights drawn from experimental results. For training the TW-MLM, we use the set of 2023 terminal walks generated from 100 Physics equations (Udrescu & Tegmark, 2020). These equations are suitable as they contain widely accepted expressions used in real-life scenarios, in contrast to the other SR datasets. When tokenizing the equations, variables are represented as a single "variable token". When TW-MLM generates a new equation for a numerical dataset, this token is replaced by a randomly selected variable.

Throughout this paper, we use a lightweight RNN, comprising of a single embedding layer, a single long short-term memory layer and a single dense layer. We experiment with replacing the RNN with a transformer, which we find to perform worse. As seen in Table 2, our experiments show that with just the sole usage of our TW-MLM to generate candidate expressions, we can drastically improve both *top-1 exact recovery rate* and *top-1 approximate recovery rate* across all 3 datasets. The intuition for the improvement of results is that the TW-MLM learns the intrinsic patterns which makes equations human-readable, which promotes the GP algorithm to explore a search space with a high concentration of human-readable equations.

## 3.3 ADAPTABILITY IN EVOLUTION

Borrowing the idea of adaptability from the theory of evolution (Bateson, 2017), we argue that a candidate expression close to the ground truth will survive through a multitude of fitness functions, whereas a pseudo-expression may perform deceptively well for one but perform worse for the other fitness measures. In this context, the challenge for SR is to find suitable secondary fitness functions that measure characteristics beyond the primary fitness function, i.e., faithfulness to empirical values via the MSE. One must also optimize for both fitness functions and we do this by alternating between the primary and secondary fitness functions.

Additionally, we note that one benefit of the simple closed-form analytical expression found by SR is that it allows the user to apply traditional mathematical tools on the expression, such as derivatives. In this paper, we explore the derivative as part of the secondary fitness functions, e.g., faithfulness to empirical derivatives. We describe two natural approaches to doing this, mean squared error of derivatives and mean squared error of difference, and argue that the former is superior to the latter for our application.

**Mean Squared Error (MSE) of Derivative (MSEDE).** MSE is commonly used as a fitness and optimisation function, and yields the Maximum Likelihood Estimate (MLE): $\theta_{MSE} = \arg\min_\theta \sum_{i=1}^{N}(y_i - \hat{y}_i)^2$. MSE rewards expressions that are faithful to the values in the datasets. Likewise, we can reward expressions for being faithful to empirical derivatives. We can compute the MSE of the empirical derivatives from the dataset and candidate expressions. Consider $N$ pairs

of values $(x_i, y_i)$, where $i = 1, 2, \ldots, N$, sorted based in ascending values of $x$. We define the empirical derivative as $\frac{\Delta y_i}{\Delta x_i} = \frac{y_{i+1} - y_i}{x_{i+1} - x_i}$. Then, $\theta_{MSEDE} = \arg\min_\theta \sum_{i=1}^{N-1} (\frac{\Delta y_i}{\Delta x_i} - \frac{\Delta \hat{y}_i}{\Delta x_i})^2$.

**Mean Squared Error of Difference (MSEDI).** Here, we develop a new fitness measure derived from a theoretical basis that is consistent with the traditional error modelling framework used in MSE: Given measurements $y = f(x) + C + \epsilon$, where C is a an arbitrary constant and $\epsilon \sim \mathcal{N}(0, \sigma)$, we aim to find $\hat{y} = g_\theta(x)$, such that $g_\theta(x)$ is a close approximation to $f(x)$. We derive the MLE of parameters $\theta$ by considering empirical derivatives.

When $g_\theta(x) \approx f(x) + C$, $\frac{\Delta y_i}{\Delta x_i} - \frac{\Delta \hat{y}_i}{\Delta x_i} = \frac{\epsilon_{i+1} - \epsilon_i}{\Delta x_i} \sim \mathcal{N}(0, \frac{2\sigma^2}{(\Delta x_i)^2})$. We note that the difference between derivatives obtained from dataset and candidate expression follows a Gaussian distribution as well, which are independent to each other under the set of odd-valued or even-valued $i$. The total log likelihood across both sets should be similar by symmetry. We choose to evaluate the total log likelihood across all $i$ instead of picking one set. By letting $\sigma_i' = \frac{\sqrt{2}\sigma}{\Delta x_i}$, the log likelihood of a particular $\frac{\Delta y_i}{\Delta x_i}$ is $\ln(Pr(\frac{\Delta y_i}{\Delta x_i}|\theta)) = -\ln(\sqrt{2\pi}\sigma_i') - \frac{1}{2\sigma_i'^2}(\frac{\Delta y_i}{\Delta x_i} - \frac{\Delta \hat{y}_i}{\Delta x_i})^2$.

The total log likelihood across all $i$ is thus,

$$\sum_{i=1}^{N-1} \left( -\ln(\sqrt{2\pi}\sigma_i') - \frac{1}{2\sigma_i'^2} \left( \frac{\Delta y_i}{\Delta x_i} - \frac{\Delta \hat{y}_i}{\Delta x_i} \right)^2 \right). \tag{1}$$

The parameters $\theta$ of $g_\theta(x)$ are obtained by maximizing the total log likelihood in (1) to obtain:

$$\theta_{MSEDI} = \arg\max_\theta \sum_{i=1}^{N-1} \left( -\ln(\sqrt{2\pi}\sigma_i') - \frac{1}{2\sigma_i'^2} \left( \frac{\Delta y_i}{\Delta x_i} - \frac{\Delta \hat{y}_i}{\Delta x_i} \right)^2 \right) \tag{2}$$

$$= \arg\max_\theta \sum_{i=1}^{N-1} \left( -\frac{1}{2\sigma_i'^2} \left( \frac{\Delta y_i}{\Delta x_i} - \frac{\Delta \hat{y}_i}{\Delta x_i} \right)^2 \right) \tag{3}$$

$$= \arg\min_\theta \sum_{i=1}^{N-1} (\Delta y_i - \Delta \hat{y}_i)^2 \tag{4}$$

**MSEDI more relevant for real-world compared to MSEDE.** However, we discover that MSEDE is a potentially harmful fitness function when dealing real-life datasets since it overfits to a selected set of noisy derivative values. Our experiments that include noise yield that using MSEDE led to no recovery across 100 repetitions of experimentation, while MSEDI was successful. We observe that random error, $\epsilon$, contributes to large noise in empirical derivatives, especially when $\Delta x_i$ is small. In other words, MSEDE can be viewed as a weighted version of MSEDI, with the weights being $\frac{1}{(\Delta x_i)^2}$. The MSEDE function thus encourages over-fitting to values which are heavily weighted. Additionally, MSEDE relies on the value $\frac{1}{\Delta x_i}$, which is problematic when $\Delta x_i$ is 0, which is common in real-world datasets with duplicated $x$ values. Thus MSEDI better fulfills the criteria of relevancy to real-life problem (White et al., 2013), providing an additional justification to choose our theoretically derived MSEDI.

**Multi-objective optimization using MSEDI and MSE.** As discussed in Section 2, it is difficult to create a general method to optimize for both fitness functions using traditional methods. For instance, using a weighted combination do not work for some equations as one fitness will dominate, effectively only optimizing for one fitness in those cases. Instead, using the idea of adaptability, we utilize MSEDI as a secondary test with the intention to filter away pseudo-expressions as described earlier in Section 3.3. Our experiments show the addition of this adaptability mechanism (using MSEDI as fitness function) once every 5 generations improves the exact recovery by more than double while maintaining a similar approximate recovery rate.

**Ablation: Escaping local optima with and without MSEDI.** In addition to acting as a secondary fitness function to remove pseudo-expressions, we also find that MSEDI functions by helping GP escape poor local optima. Among the experiments that do not recover the equation, we save the state of the GP and run it for an additional 5 generations of MSE. If the top-1 equation remains the same (which occurred for 72.82% of experiments), we rollback the state of the GP and run just 1 generation of MSEDI. The percentage of experiments in which this 1 generation of MSEDI helps

to change the top equation are 19.82%, 15.58%, 44.33% for the *Nguyen\**, *R\** and *Jin\** datasets respectively. Thus, from the results we know that MSEDI helps to escape poor local optima which would have otherwise stagnated.

## 4 INTEGRATED PROOF-OF-CONCEPT

Combining the ideas described above, we added TW-MLM and the adaptability mechanism, that alternates MSE and MSEDI as fitness function, to GP with constant optimization and using the primitive function set $\{add, mul, sub, div, sin, cos, arcsin, log, exp, pow\}$ as discussed in Section 3.1. The interfacing of each component can be visualized in a simple schematic (Figure 1).

---

**Algorithm 1:** Outline of our Integrated Proof-of-Concept

---

**while** *current generation < max generations* **do**

    Fill up population of equations by generating equations from TW-MLM;

    **if** *current generation % 5* **then**

        Evaluate fitness of equations using MSEDI

    **else**

        Evaluate fitness of equations using MSE

    **end**

    Evolve and select equations based on evolutionary operations in GP (e.g., crossover, subtree mutation, hoist mutation, point mutation, reproduction)

**end**

---

Algorithm 1 outlines our proof-of-concept. TW-MLM is trained once on a MLM dataset and utilized to generate candidate expressions at the start of every generation of GP. GP then evaluates and filters from this set of expressions together with candidates derived from the previous generation through a fitness functions that alternates between generation. In our method, we use MSE as our base fitness function and switch to MSEDI once every 5 generations (decided based on hyper-parameter tuning on a smaller-sized experiment). We then conduct a series of experiments to test the proposed approach on both *synthetic* and *real-world datasets*.

**Synthetic Datasets Experiments.** We compare with the benchmarks methods and random search outlined in Section 2. We choose synthetic datasets *with unknown constants* (i.e., *Jin\**) and *without unknown constants* (i.e., *Nguyen\** and *R\**). Each equation in Table 6 (Appendix) is used to conduct 100 experiments per method. In Table 3 and Table 4, we tabulate the performance on the three synthetic datasets (which contain 17 equations to be recovered).

**Synthetic Datasets Performance Comparison.** Our method shows overall competitive performance with state-of-the-art. On equations with unknown constants, shown in the *Jin\** dataset in Table 3, our method outperforms all other methods by a large margin, in both *top-1 approximate recovery rate* and *top-1 exact recovery rate*. On equations without unknown constants (e.g., *Nguyen\**), our method outperforms all except DSO-NGGP for almost all of the equations, as shown in Table 4. Since *Jin\** included unknown constants that allows for relations that are reflective of real-life

Table 3: *Synthetic Dataset:* Top-1 Approximate / Top-1 Exact Recovery Rates (%) of 6 methods for *Jin\** dataset. Results were averaged over 100 experiments per equation per method.

|  |  | OURS | RANDOM SEARCH | TURINGBOT | GPLEARN | DSR | DSO-NGGP |
|---|---|---|---|---|---|---|---|
| JIN* | JIN*-1 | **78/3** | 0/0 | 1/0 | 29/0 | 10/0 | 47/0 |
|  | JIN*-2 | **58/25** | 0/0 | 1/0 | 23/0 | 0/0 | 2/0 |
|  | JIN*-3 | **53/7** | 0/0 | 0/0 | 16/0 | 0/0 | 12/0 |
|  | JIN*-4 | **31/10** | 0/0 | 0/0 | 21/0 | 0/0 | 0/0 |
|  | JIN*-5 | **21/20** | 1/0 | 1/0 | 14/3 | 0/0 | 7/5 |
|  | JIN*-6 | 0/0 | 0/0 | 0/0 | 0/0 | 0/0 | 0/0 |
|  | **AVERAGE** | **40.17/10.83** | 0.17/0.00 | 0.50/0.00 | 17.16/0.50 | 1.67/0.00 | 11.33/0.83 |

Table 4: *Synthetic Dataset:* Top-1 Approximate / Top-1 Exact Recovery Rates (%) of 6 methods for *Nguyen\** and *R\** datasets. Results were averaged over 100 experiments per equation per method.

|  | OURS | RANDOM SEARCH | TURINGBOT | GPLEARN | DSR | DSO-NGGP |
|---|---|---|---|---|---|---|
| **NGUYEN*-AVERAGE** | 42.13/18.38 | 17.63/0.00 | 40.50/1.88 | 19.00/2.00 | 30.63/7.13 | 68.25/23.38 |
| **R*-AVERAGE** | 21.00/0.00 | 0.67/0.00 | 6.00/0.00 | 5.33/0.00 | 0.67/0.00 | 23.67/0.00 |

Table 5: *Real-World Dataset (from SRBench):* Percentage of experiments where our method outperforms DSO-NGGP in r-squared value. Results are shown for 100 experiments per dataset.

| NAME OF DATASET | 1027_ESL | 1028_SWD | 1029_LEV | 1030_ERA | STROGATZ_SHEARFLOW1 | 192_VINEYARD_SMALL |
|---|---|---|---|---|---|---|
| OUTPERFORMANCE | 57% | 66% | 88% | 94% | 73% | 78% |

variables such as feature scaling (Udrescu & Tegmark, 2020), it can be argued that the results from *Jin\** hold more practical value. In this sense, our model clearly outperformed all models as seen in Table 3.

**Real-World Datasets (SRBench) Evaluation.** We then evaluated our method against DSO-NGGP on 6 real-world datasets from SRBench (La Cava et al., 2021), with the results tabulated in Table 5. Unlike traditional SR datasets, real-world datasets do not have an accompanying ground truth equation. We present the percentage of experiments in which our method outperformed DSO-NGGP in terms of r-squared value. Our method consistently outperforms DSO-NGGP, further corroborating our previous results and strengthening the claim that SR datasets with unknown constants are more reflective of real-world dataset performance.

**Intuition and Ablation.** The function of TW-MLM in our method is to implicitly learn the rules and constraints about the morphology of human-readable equations that are previously developed based on human judgment. This increases the likelihood of our method to venture into the search space containing expressions that are consistent with existing widely-accepted human-readable equations. In this sense, TW-MLM acts as a guide in the vast search space of possible equations. Though man-made heuristics such as rules and constraints exist to fulfill these roles, it is difficult to find, difficult to express and not easily transferable to new problems. On the other hand, TW-MLM finds these rules and constraints implicitly and the method can be easily transferred to a new problem. The adaptability mechanism (by alternating between MSE and MSEDI fitness function) then synergizes with TW-MLM and GP by increasing the difficulty for pseudo-equations to survive each generation, as shown in the ablation study in Section 3.2. We also observed throught the ablation study in Section 3.3 how MSEDI helps to escape from poor optima points in GP. These mechanisms added to GP allow us to create a model that performs competitively with state-of-the-art.

## 5 REFLECTIONS

**Summary.** In this paper, we demonstrate an efficient proof-of-concept that incorporates two new independent mechanisms into genetic programming-based SR: (a) Terminal walks mathematical language model (TW-MLM) and (b) Adaptability via alternating fitness functions (i.e., between MSE and our novel theoretically justified metric called MSEDI). Through these simple modifications, we are able to obtain competitive results with respect to a diverse range of methods, outperforming all, including state-of-the-art, when datasets with unknown constants are involved. We then reproduce this outperformance on real-world datasets. We also state and justify the conventions we use that can promote consistency, comparability and relevance to real-world problems. Ultimately, we hope that SR can demonstrate more competitive real-world results among other machine learning methods, given its natural advantage in both interpretability and explainability. The code for this paper is available at: https://github.com/kentridgeai/MorphologyAndAdaptabilitySR

**Limitations.** Our method performs much better on equations with unknown constants (*Jin\**), in contrast to equations without unknown constants (*Nguyen\**). The reason behind this observation can be attributed to the difference in frequency of unknown constants during the generation of candidate expressions. Our TW-MLM is trained on 100 AI-Feynman equations which has the frequent use of constants. This means that the TW-MLM has to learn to include constants into candidate expressions frequently, increasing the chances of over-optimization of constants in candidate expressions.

**Future Work.** (i) The performance of our method on datasets without unknown constants may suffer due to over-optimization of constants in candidate expressions. To reduce the chances of this, SR can alternate between randomly generating constants and optimizing for constants. This is similar to the way we alternate between fitness functions in this paper. (ii) The search space of SR grows rapidly with an increasing number of variables, making real-world datasets with many variables a difficult problem for SR. We are currently working on an iterative greedy approach to multi-variable SR, dealing with one additional variable per SR-run to address the issues that SR faces with time complexity.

ACKNOWLEDGMENTS

This research is supported by A*STAR, CISCO Systems (USA) Pte. Ltd and the National University of Singapore under its Cisco-NUS Accelerated Digital Economy Corporate Laboratory (Award I21001E0002). Additionally, we would like to thank the members of the Kent-Ridge AI research group at the National University of Singapore for helpful feedback and interesting discussions on this work.

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

# A    APPENDIX

Table 6: Symbolic regression dataset specifications. Input variable is denoted by $x$. $U$(a, b, c) denotes c random points uniformly sampled between a and b for $x$. Equations were selected or modified to have the same number of variables throughout to maintain the same search space size to allow for meaningful comparison.

| NAME | EXPRESSION | SAMPLING RANGE |
|------|-----------|----------------|
| NGUYEN*-1 | $x^3 + x^2 + x$ | $U(-1, 1, 20)$ |
| NGUYEN*-2 | $x^4 + x^3 + x^2 + x$ | $U(-1, 1, 20)$ |
| NGUYEN*-3 | $x^5 + x^4 + x^3 + x^2 + x$ | $U(-1, 1, 20)$ |
| NGUYEN*-4 | $x^6 + x^5 + x^4 + x^3 + x^2 + x$ | $U(-1, 1, 20)$ |
| NGUYEN*-5 | $\sin\left(x^2\right)\cos(x) - 1$ | $U(-1, 1, 20)$ |
| NGUYEN*-6 | $\sin(x) + \sin\left(x + x^2\right)$ | $U(-1, 1, 20)$ |
| NGUYEN*-7 | $\log(x + 1) + \log\left(x^2 + 1\right)$ | $U(0, 2, 20)$ |
| NGUYEN*-8 | $\sqrt{x}$ | $U(0, 4, 20)$ |
| R*-1 | $\frac{(x+1)^3}{x^2 - x + 1}$ | $U(-1, 1, 20)$ |
| R*-2 | $\frac{x^5 - 3x^3 + 1}{x^2 + 1}$ | $U(-1, 1, 20)$ |
| R*-3 | $\frac{x^6 + x^5}{x^4 + x^3 + x^2 + x + 1}$ | $U(-1, 1, 20)$ |
| JIN*-1 | $2.5x^4 - 1.3x^3 + 0.5x^2 - 1.7x$ | $U(-3, 3, 100)$ |
| JIN*-2 | $8.0x^3 + 8.0x^2 - 15.0$ | $U(-3, 3, 100)$ |
| JIN*-3 | $0.7x^3 - 1.7x$ | $U(-3, 3, 100)$ |
| JIN*-4 | $1.5\exp(x) + 5.0\cos(x)$ | $U(-3, 3, 100)$ |
| JIN*-5 | $6.0\sin(x)\cos(x)$ | $U(-3, 3, 100)$ |
| JIN*-6 | $1.35x^2 + 5.5\sin\left((x - 1.0)^2\right)$ | $U(-3, 3, 100)$ |

Table 7: 100 AI-Feynman Physics Equations Udrescu & Tegmark (2020)

| Feynman eq. | Equation |
|---|---|
| I.6.20a | $f = e^{-\theta^2/2}/\sqrt{2\pi}$ |
| I.6.20 | $f = e^{-\frac{\theta^2}{2\sigma^2}}/\sqrt{2\pi\sigma^2}$ |
| I.6.20b | $f = e^{-\frac{(\theta-\theta_1)^2}{2\sigma^2}}/\sqrt{2\pi\sigma^2}$ |
| I.8.14 | $d = \sqrt{(x_2-x_1)^2 + (y_2-y_1)^2}$ |
| I.9.18 | $F = \frac{Gm_1m_2}{(x_2-x_1)^2+(y_2-y_1)^2+(z_2-z_1)^2}$ |
| I.10.7 | $m = \frac{m_0}{\sqrt{1-\frac{v^2}{c^2}}}$ |
| I.11.19 | $A = x_1y_1 + x_2y_2 + x_3y_3$ |
| I.12.1 | $F = \mu N_n$ |
| I.12.2 | $F = \frac{q_1q_2}{4\pi\epsilon r^2}$ |
| I.12.4 | $E_f = \frac{q_1}{4\pi\epsilon r^2}$ |
| I.12.5 | $F = q_2 E_f$ |
| I.12.11 | $F = q\left(E_f + Bv\sin\theta\right)$ |
| I.13.4 | $K = \frac{1}{2}m\left(v^2 + u^2 + w^2\right)$ |
| I.13.12 | $U = Gm_1m_2\left(\frac{1}{r_2} - \frac{1}{r_1}\right)$ |
| I.14.3 | $U = mgz$ |
| I.14.4 | $U = \frac{k_{spring}x^2}{2}$ |
| I.15.3x | $x_1 = \frac{x-ut}{\sqrt{1-u^2/c^2}}$ |
| I.15.3t | $t_1 = \frac{t-ux/c^2}{\sqrt{1-u^2/c^2}}$ |
| I.15.10 | $p = \frac{m_0v}{\sqrt{1-v^2/c^2}}$ |
| I.16.6 | $v_1 = \frac{u+v}{1+uv/c^2}$ |
| I.18.4 | $r = \frac{m_1r_1+m_2r_2}{m_1+m_2}$ |
| I.18.12 | $\tau = rF\sin\theta$ |
| I.18.16 | $L = mrv\sin\theta$ |
| I.24.6 | $E = \frac{1}{4}m\left(\omega^2 + \omega_0^2\right)x^2$ |
| I.25.13 | $V_e = \frac{q}{C}$ |
| I.26.2 | $\theta_1 = \arcsin\left(n\sin\theta_2\right)$ |
| I.27.6 | $f_f = \frac{1}{\frac{1}{d_1}+\frac{n}{d_2}}$ |
| I.29.4 | $k = \frac{\omega}{c}$ |
| I.29.16 | $x = \sqrt{x_1^2 + x_2^2 - 2x_1x_2\cos\left(\theta_1-\theta_2\right)}$ |
| I.30.3 | $I_* = I_{*0}\frac{\sin^2(n\theta/2)}{\sin^2(\theta/2)}$ |
| I.30.5 | $\theta = \arcsin\left(\frac{\lambda}{nd}\right)$ |
| I.32.5 | $P = \frac{q^2a^2}{6\pi\epsilon c^3}$ |
| I.32.17 | $P = \left(\frac{1}{2}\epsilon c E_f^2\right)\left(8\pi r^2/3\right)\left(\omega^4/\left(\omega^2-\omega_0^2\right)^2\right)$ |
| I.34.8 | $\omega = \frac{qvB}{p}$ |
| I.34.10 | $\omega = \frac{\omega_0}{1-v/c}$ |
| I.34.14 | $\omega = \frac{1+v/c}{\sqrt{1-v^2/c^2}}\omega_0$ |
| I.34.27 | $E = \hbar\omega$ |
| I.37.4 | $I_* = I_1 + I_2 + 2\sqrt{I_1 I_2}\cos\delta$ |
| I.38.12 | $r = \frac{4\pi\epsilon\hbar^2}{mq^2}$ |
| I.39.10 | $E = \frac{3}{2}p_F V$ |
| I.39.11 | $E = \frac{1}{\gamma-1}p_F V$ |
| I.39.22 | $P_F = \frac{nk_bT}{V}$ |
| I.40.1 | $n = n_0 e^{-\frac{mgx}{k_bT}}$ |

| Feynman eq. | Equation |
|---|---|
| I.41.16 | $L_{rad} = \frac{\hbar \omega^3}{\pi^2 c^2 \left( e^{\frac{\hbar \omega}{k_b T}} - 1 \right)}$ |
| I.43.16 | $v = \frac{\mu_{drift} q V_e}{d}$ |
| I.43.31 | $D = \mu_e k_b T$ |
| I.43.43 | $\kappa = \frac{1}{\gamma - 1} \frac{k_b v}{A}$ |
| I.44.4 | $E = n k_b T \ln \left( \frac{V_2}{V_1} \right)$ |
| I.47.23 | $c = \sqrt{\frac{\gamma p r}{\rho}}$ |
| I.48.20 | $E = \frac{m c^2}{\sqrt{1 - v^2/c^2}}$ |
| I.50.26 | $x = x_1 \left[ \cos(\omega t) + \alpha \cos(\omega t)^2 \right]$ |
| II.2.42 | $P = \frac{\kappa (T_2 - T_1) A}{d}$ |
| II.3.24 | $F_E = \frac{P}{4 \pi r^2}$ |
| II.4.23 | $V_e = \frac{q}{4 \pi \epsilon r}$ |
| II.6.11 | $V_e = \frac{1}{4 \pi \epsilon} \frac{p_d \cos \theta}{r^2}$ |
| II.6.15a | $E_f = \frac{3}{4 \pi \epsilon} \frac{p_d z}{r^5} \sqrt{x^2 + y^2}$ |
| II.6.15b | $E_f = \frac{3}{4 \pi \epsilon_d} \frac{p_d}{r^3} \cos \theta \sin \theta$ |
| II.8.7 | $E = \frac{3}{5} \frac{q^2}{4 \pi \epsilon d}$ |
| II.8.31 | $E_{\text{den}} = \frac{\epsilon E_f^2}{2}$ |
| II.10.9 | $E_f = \frac{\sigma_{\text{den}}}{\epsilon} \frac{1}{1 + \chi}$ |
| II.11.3 | $x = \frac{q E_f}{m (\omega_0^2 - \omega^2)}$ |
| II.11.17 | $n = n_0 \left( 1 + \frac{p_d E_f \cos \theta}{k_b T} \right)$ |
| II.11.20 | $P_* = \frac{n_\rho p_d^2 E_f}{3 k_b T}$ |
| II.11.27 | $P_* = \frac{n \alpha}{1 - n \alpha/3} \epsilon E_f$ |
| II.11.28 | $\theta = 1 + \frac{n \alpha}{1 - (n \alpha/3)}$ |
| II.13.17 | $B = \frac{1}{4 \pi \epsilon c^2} \frac{2 I}{r}$ |
| II.13.23 | $\rho_c = \frac{\rho_{c_0}}{\sqrt{1 - v^2/c^2}}$ |
| II.13.34 | $j = \frac{\rho_{c_0} v}{\sqrt{1 - v^2/c^2}}$ |
| II.15.4 | $E = -\mu_M B \cos \theta$ |
| II.15.5 | $E = -p_d E_f \cos \theta$ |
| II.21.32 | $V_e = \frac{q}{4 \pi \epsilon r (1 - v/c)}$ |
| II.24.17 | $k = \sqrt{\frac{\omega^2}{c^2} - \frac{\pi^2}{d^2}}$ |
| II.27.16 | $F_E = \epsilon c E_f^2$ |
| II.27.18 | $E_{den} = \epsilon E_f^2$ |
| II.34.2a | $I = \frac{q v}{2 \pi r}$ |
| II.34.2 | $\mu_M = \frac{q v r}{2}$ |
| II.34.11 | $\omega = \frac{g_- q B}{2 m}$ |
| II.34.29a | $\mu_M = \frac{q h}{4 \pi m}$ |
| II.34.29b | $E = \frac{g_- \mu_M B J_z}{\hbar}$ |
| II.35.18 | $n = \frac{n_0}{\exp(\mu_m B/(k_b T)) + \exp(-\mu_m B/(k_b T))}$ |
| II.35.21 | $M = n_\rho \mu_M \tanh \left( \frac{\mu_M B}{k_b T} \right)$ |
| II.36.38 | $f = \frac{\mu_m B}{k_b T} + \frac{\mu_m \alpha M}{\epsilon c^2 k_b T}$ |
| II.37.1 | $E = \mu_M (1 + \chi) B$ |
| II.38.3 | $F = \frac{Y A x}{d}$ |
| II.38.14 | $\mu_S = \frac{Y}{2(1 + \sigma)}$ |
| III.4.32 | $n = \frac{1}{e^{\frac{\hbar \omega}{k_b T}} - 1}$ |

| Feynman eq. | Equation |
|---|---|
| III.4.33 | $E = \frac{\hbar\omega}{\frac{\hbar\omega}{k_bT}-1}$ |
| III.7.38 | $\omega = \frac{2\mu_M B}{\hbar}$ |
| III.8.54 | $p_\gamma = \sin\left(\frac{Et}{\hbar}\right)^2$ |
| III.9.52 | $p_\gamma = \frac{p_d E_f t}{\hbar} \frac{\sin((\omega-\omega_0)t/2)^2}{((\omega-\omega_0)t/2)^2}$ |
| III.10.19 | $E = \mu_M \sqrt{B_x^2 + B_y^2 + B_z^2}$ |
| III.12.43 | $L = n\hbar$ |
| III.13.18 | $v = \frac{2Ed^2 k}{\hbar}$ |
| III.14.14 | $I = I_0 \left(e^{\frac{qV_e}{k_bT}} - 1\right)$ |
| III.15.12 | $E = 2U(1 - \cos(kd))$ |
| III.15.14 | $m = \frac{\hbar^2}{2Ed^2}$ |
| III.15.27 | $k = \frac{2\pi\alpha}{nd}$ |
| III.17.37 | $f = \beta(1 + \alpha\cos\theta)$ |
| III.19.51 | $E = \frac{-mq^4}{2(4\pi\epsilon)^2\hbar^2}\frac{1}{n^2}$ |
| III.21.20 | $j = \frac{-\rho_0 q A_{\text{vec}}}{m}$ |

