# OpenReview forum: "Rethinking Symbolic Regression: Morphology and Adaptability in the Context of Evolutionary Algorithms"
_ICLR.cc/2023/Conference — ICLR 2023 poster_

### Official Review · Reviewer_Kqo2 · 2022-10-21

**Confidence:** 4
**Correctness:** 3
**Technical Novelty And Significance:** 3
**Empirical Novelty And Significance:** 3
**Recommendation:** 8

**Clarity, Quality, Novelty And Reproducibility:**

Quality:

This paper is of good quality overall as an empirical study. It is well motivated, clearly written, and well supported by experiments. However, the theoretical justification is not solid, as I stated in the weakness part.

Clarity:

This paper is clearly written and properly related to the literature.

Originality:

This paper combines two existing ideas in other fields, but the combination itself is normal.

**Details Of Ethics Concerns:**

I have no ethics concerns.

**Strength And Weaknesses:**

Strengths:

1. A novel combination of existing techniques.

This paper presents a novel combination of existing techniques in other areas for solving the symbolic regression problem. Although these ideas already exist, the important point is to identify their roles for symbolic regression and properly combine them to achieve better performance in symbolic regression.

2. Clearly stated ideas.

The ideas in the proposed approach are clearly stated. In particular, I appreciate that they are properly related to their original literature, and why it makes sense to use these techniques is well explained.

3. Supporting experiments.

Another strength of this paper is the experiments that support the arguments and deliver practical messages. For example, it is good to realize that different methods may perform differently with and without unknown constants, and recognize the importance of unknown constants in practice.



Weaknesses:

1. Theoretical justification.

The only theoretical derivation on page 7 looks suspicious and I am not convinced. To be specific, even if the gaussian model is correct, the likelihood function is incorrect: $\epsilon_{i+1}-\epsilon_i$ are not i.i.d. random variables, hence the likelihood function cannot be a product over $i=1,\dots,n$. Although MSEDI performs well in the experiments, the authors still need to think carefully about how to justify this criterion.

2. Reproducibility.

As this paper is mostly an empirical study, the current version falls short due to the lack of open-source reproduction codes or libraries. I understand that this may be restricted in the blinded version. But at least it should be provided later on.

3. The choice of RNN architecture. (question)

This point is more of a question: how do you choose the RNN architecture for your language model? Did you do some selection based on their performance, or are they commonly used in the area of language models?

**Summary Of The Paper:**

This paper combines ideas in language modeling and the theory of evolution to develop a novel approach to symbolic regression. By incorporating the terminal walk in language modeling, the proposed method is argued and shown to be able to reduce the reliance on human heuristics for constraints on the learned expression. By incorporating adaptivity from the theory of evolution, the proposed method is shown to be able to accommodate various learning targets, and perform better especially in practical situations when the true expression contains unknown parameters. The ideas are clearly stated and related literatures are properly compared to. The arguments are also backed by numerical experiments.

**Summary Of The Review:**

I find this to be a nice paper. It presents a novel approach that combines ideas from language models and evolution theory to solve symbolic regression. It properly relates to the literature, clearly states the benefits and intuitions of using these ideas, and supports these arguments by numerical experiments, which further identify the importance and challenges of accounting for unknown parameters. However, the authors need to be careful about the theoretical justification, and should provide reproduction codes when possible.

---

> ### Author Response · Authors · 2022-11-19
> **Response to reviewer Kqo2**
>
> We thank reviewer Kqo2 for the valuable feedback and recommendations. We address the main concerns below:
>
> 1. “Theoretical justification”
>
> We thank the reviewer for the comment. Under the set of all odd-valued i, $ϵ_{i+1}$−$ϵ_{i}$ are i.i.d random variables. The same applies for the set of all even-valued i. Since there is no substantial difference between choosing odd or even valued i, we maximize for the product of the likelihood over the 2 sets, allowing us to obtain MSEDI. We have revised the paper to clarify this point.
>
> 2. “Reproducibility”
>
> We thank the reviewer for the comment.  In response, we foreshadow our proof-of-concept in greater detail starting right from the introduction and add an overall algorithm flowchart. Also, we have updated the method description to be clearer and provided the architecture of the layers in the neural network. We have also provided implementation code with settings in the revision.
>
> 3. “The choice of RNN architecture.”
>
> We thank the reviewer for the suggestion. RNNs and Transformers are commonly used in language modeling tasks. As such, we initially provided results using the RNN architecture (details and code are now provided). We have also revised the paper to include results of a transformer model in Table 2 as part of our ablation study. We selected the RNN architecture due to a slight outperformance over the transformer. Nonetheless, the increase of the performance of the transformer over vanilla GP demonstrates that the effectiveness of TW-MLM is not overly reliant on the architecture, as shown by the results.

---

### Official Review · Reviewer_yTKT · 2022-10-25

**Confidence:** 4
**Correctness:** 3
**Technical Novelty And Significance:** 3
**Empirical Novelty And Significance:** 3
**Recommendation:** 6

**Clarity, Quality, Novelty And Reproducibility:**

The paper is well written and well structured. The authors provide clear treatment of the area of contribution and positioning of the proposed approach. The novelty lies in the proposed mathematical language model and its combination with standard GP in a multi optimization context. The methods are well described and can be potentially reproduced.



**Strength And Weaknesses:**

Strengths:
- Well argumented
- Well evaluated
- Contributions both to methods for solving SR and evaluation/benchmarking of SR methods
Weaknesses:
- Ambitious goal of setting conventions
- Conventions are
- Unnecessary introduction of novel terminology

**Summary Of The Paper:**

The authors propose an evolutionary algorithm for symbolic regression combining genetic programming and a mathematical language model. The authors propose reasonable conventions that they use in their evaluation and suggest to be used by the community in general. To generate examples for training a lightweight RNN mathematical language model they propose to use a terminal instead of prefix representation. To solve the task of SR they also propose a method for alternating fitness functions for multi optimization. They evaluate their approach on synthetic datasets with and without unknown constants and compare it to a selection of SR approaches. Additionally they compare their approach to DSO-NGGP on a real-world datasets from SRBench.

**Summary Of The Review:**

Overall a good, well written paper. The narrative can be slightly toned down. While the suggestions for evaluation conventions may be ultimately of benefit to the community a broader review and discussion is needed on this topic. While it is good to propose conventions and learn from the best practices, here the proposed is an evaluation frame used to demonstrate the performance of the presented approach. I believe that it is unnecessary to introduce novel terminology for concepts that are well known and understood such as morphology for structure of the model and adaptability for objective criteria. The authors themselves establish 1-to-1 relationship between current terminology and their new terminology as early as in the abstract and the introduction section.

Regarding the proposed top-1 approximate recovery, i agree that perhaps the best scoring solution should be taken into account when comparing methods. This, however, removes the ability to evaluate one major performance aspect of SR methods - the consistency of recovery of expressions. A good SR method should consistently produce (symbolically or goodness-of-fit) good solutions across runs. Additionally, using only a goodness-of-fit criterion (r-squared), without considering symbolic equivalency can lead to overfitted expressions and/or expressions with spurious variable interactions that can't be easily explained. This in turn also reduces the compactness and interpretability of the solutions, which is a major upside of SR methods over other black-box methods. At least the parsimony of generated expressions should not be underestimated especially in the context of scientific discovery. I accept that this might not be exactly the case and I have misunderstood the proposed convention. In that case it would be good that authors explain this concept additionally and remove concerns by providing an empirical demonstration at least in the scope of their evaluation.

Regarding the proposed selection of appropriate lifetime population size, this applies mostly to population-based algorithms. Additionally, the authors propose a selection of arbitrary population size or a heuristic for selecting a population size based on the performance of a  random search (may be out of scope for this work, but the authors should take a look at Bergrstra and Bengio, JMLR, 2012 on random search). Making the selection of arbitrary lifetime population size a convention can lead to biased evaluation, enabling selection of sizes where preferred approaches show favorable results. This also casts a bit of doubt on the presented evaluation of the approach. In fact, as it is the case in related work, instead of comparison based on a single point, a more general picture of the performance of an approach can be obtained by looking at the performance as a function of the number of evaluations instead. This would strengthen the performance claims of the proposed approach and as a convention it would also be more inclusive to other, non-evolutionary approaches.

---

> ### Author Response · Authors · 2022-11-19
> **Response to reviewer yTKT's feedback**
>
> We thank reviewer yTKT for the valuable feedback and recommendations. We address the main concerns below:
>
> 1. “Regarding the proposed top-1 approximate recovery, i agree that perhaps the best scoring solution should be taken into account when comparing methods. This, however, removes the ability to evaluate one major performance aspect of SR methods - the consistency of recovery of expressions. A good SR method should consistently produce (symbolically or goodness-of-fit) good solutions across runs. Additionally, using only a goodness-of-fit criterion (r-squared), without considering symbolic equivalency can lead to overfitted expressions and/or expressions with spurious variable interactions that can't be easily explained.”
>
> We thank the reviewer for the pointers. We also value consistency of recovery of expressions, and the method we have chosen to evaluate consistency is by evaluating 100 trials per dataset. Also, we agree that symbolic equivalence can still be informative about the performance of the methods, and hence we decided to include the recovery percentage in brackets in our paper.
>
> 2. “Regarding the proposed selection of appropriate lifetime population size, this applies mostly to population-based algorithms. Additionally, the authors propose a selection of arbitrary population size or a heuristic for selecting a population size based on the performance of a random search (may be out of scope for this work, but the authors should take a look at Bergrstra and Bengio, JMLR, 2012 on random search). Making the selection of arbitrary lifetime population size a convention can lead to biased evaluation, enabling selection of sizes where preferred approaches show favorable results.”
>
> We thank the reviewer for the pointer. The practical motivation behind this was that previous experiments with a human-selected large population size led to extremely good performance of the SR methods. However, when evaluated on a random search method, we found that random search performs competitively. This made the results on the SR methods less significant. Though the selection of a population size can lead to biased evaluation, we attempt to minimize this by deciding this population on a small subset of experiments, similar to how traditional hyper parameter tuning is done.

---

> > ### Comment · Reviewer_yTKT · 2022-11-21
> > **Response**
> >
> > I would like to thank the authors for their response. In my initial evaluation I was leaning towards a more positive outcome, believing that the authors will take the opportunity to address some of the raised issues. I am a bit surprised that the authors didn't make any changes to directly address the comments and improve their work. I would like to point the authors again to my first comment in the summary of the review which they missed to even comment on. I stand behind my initial evaluation, but in case of an unclear decision i would not lean towards acceptance.

---

> > > ### Author Response · Authors · 2022-11-21
> > > **Response to reviewer yTKT's feedback Part 1 of 2**
> > >
> > > We thank the reviewer for the follow-up response. Our initial response did attempt to clarify points made in the original review (though we realize we missed the first comment in the summary of the review). As such, we would like to provide a more detailed response to the three points in the reviewer’s comments.
> > >
> > > Regarding the first point in the summary of the review:
> > >
> > > We agree on the need for a broader discussion on evaluation conventions and the proposed evaluation conventions were intended to benefit the community. With current evaluation conventions, (as we discuss in Section 3.1), recovery rates were extremely high across all methods (even for the purely random methods), which made comparisons uninformative. In fact, we even carried out mini-experiments with results to justify the proposed conventions (see results and discussion in Section 3.1). We note that the proposed conventions are independent of the proposed approach and are based on our assessment of the current evaluation landscape. We will highlight this point in the next revision.
> > >
> > > As to the use of terminology, the terms ‘morphology’ and ‘adaptability’ are used for the following reasons: (i) they are helpful to make the ideas we want to convey precise; and (ii) they are commonly used in the field of evolutionary theory (which is connected to the fact that we are using genetic programming). We use 'morphology' to refer to the 'structure of expressions' and this helps to distinguish against 'structure of the SR algorithm'. For ‘adaptability’, what we mean is the idea of an alternating fitness function (inspired by adaptation in evolutionary theory), rather than ‘objective criteria’.  We believe that the term ‘objective criteria’ does not capture the idea of ‘adaptability’; we can have an ‘objective criteria’ without having any adaptability mechanism in place. We thought that the term 'adaptability' would best capture the essence of both the mechanism and the inspiration behind the mechanism in a single word. In the next revision, we will be clearer about why we chose the terminology we did. We are, of course, open to a better way of expressing the ideas.
> > >
> > > Regarding the second point in the summary of the review:
> > >
> > > We agree that consistency of recovery is important, and we address this through evaluating the performance over 100 repeated experiments. In our evaluation, we use both top-1 approximate recovery (i.e., the best scoring solution) and top-1 exact recovery. The top-1 approximate approach (goodness of fit with r-squared metric) is relevant for real-world datasets, for which we often do not have an exact solution. We also agree that using only a ‘goodness-of-fit criterion’ may result in neglecting the potential for overfitting and neglecting the parsimony of equations. As such, we include results for both our goodness-of-fit criterion (top-1 approximate recovery) and symbolic equivalency (top-1 exact recovery) results in our paper, specifically in Table 1,2,3,4 of our revised paper, where we demonstrate empirically our findings over multiple datasets (both synthetic and real-world datasets and reporting results across 100 different trials).  In the next revision, we will fully explain the thought process behind choosing both the top-1 approximate and top-1 exact recovery metrics.

---

> > > ### Author Response · Authors · 2022-11-21
> > > **Response to reviewer yTKT's feedback Part 2 of 2**
> > >
> > > Regarding the third point in the summary of the review:
> > >
> > > We agree, to a certain extent, that the use of a fixed lifetime population size could potentially lead to biased evaluation. We also agree with the reviewer that a completely arbitrary selection of population size can allow researchers to select population sizes that show favourable results, which is precisely what we were trying to tackle. Our strategy to reduce the impact of this is to use a small subset of experiments on the random search method to set the lifetime population size. Note that we do not evaluate or tune to any other specific methods, so there is no bias towards other specific methods here. We will explain this more clearly in the next revision.
> > >
> > > Regarding the point about presenting the performance for a single-point (lifetime population size) versus as a function of a number of points, we agree that the latter would paint a more general picture of the performance of an approach. The primary reason we used a single lifetime population size was to respect the existing evaluation methods by recent state-of-the-art SR experiments. The price of the multiple evaluation points would of course be computational complexity as we would have to present the performance of the existing state-of-the-art algorithms in this manner. We hope the results (e.g., Tables 1,2,3,4) in their current form are convincing and highlight the approaches presented in this paper. We are seriously looking at the reviewer’s suggestion for multiple evaluation points as an avenue of future work.
> > >
> > > Finally, we thank the reviewer for the link to the 2012 JMLR paper on random search for hyperparameter optimization by Bergstra and Bengio. There is a clear parallel between the paper’s advocacy of random search for hyperparameter tuning and our approach for the performance of random search to set the lifetime population size. We will make the connections to the cited paper clearly in the next revision.

---

### Official Review · Reviewer_jSK3 · 2022-10-25

**Confidence:** 3
**Correctness:** 4
**Technical Novelty And Significance:** 4
**Empirical Novelty And Significance:** 4
**Recommendation:** 6

**Clarity, Quality, Novelty And Reproducibility:**

- The authors are clear about their method and indeed present a novel approach. However, the lack of information regarding the Language Model training procedure hurts reproducibility.

**Strength And Weaknesses:**

Strength:

- The experimental setup of the authors is very strong, showing different baselines and good results across different modalities of datasets.
- The ablations studies also look sound, with some really interesting findings.
- This work is also well written with no major mistakes that I could find, coupled with the right information when needed.

Weaknesses:

- (minor) I think this work would benefit a lot from a more classic structure. The authors also used their method section (Section 3) to present some ablations and results. This causes some confusion and makes the actual method not that clear. Perhaps moving the ablations afterward to the method’s actual result would clarify things. Also, the “conclusion” section, or as the authors called Reflection, seems a little too extensive for this work (will go into this later).
- (major) The authors motivate their work on the premise that deep learning methods are black-box models that remove the possibility of insights when looking into SR. However, later in the introduction the authors admit that using deep learning models is quite good for generalization propose. Their method also uses an RNN as the backbone, which to me seems that would introduce the problem of black-box models into SR again.
- (minor) Since we are using language models, I wish there were details on why not using Transformers/BERT or similar.
- (medium) The authors give no instructions on how to reproduce their work. Yes, the intuitions are there, but what is the structure of the LSTM, embedding, and dense layer they used? How many neurons are we talking about? How many hidden layers does the LSTM have? Is it a bi-lstm? Furthermore, how was this LM trained? Which optimizer? Which LR? The authors should have answered all these questions in their work, and by not doing so harm their reproducibility.
- (minor) the authors abbreviate twice MSE, MSEDE, and MSEDI
- (medium) I wish there were more in-depth explanations about the results. The authors invest a long time in the ablations with intuition and provide none in the result section of this work.
- (medium) I wish the authors had added the methods with human heuristics in their comparison. It would be ideal to have an ablation study on the trade-off from creating heuristics manually (and perhaps adding bias) and machine-made.
- (medium) If there is a concern regarding bias in man-made heuristics, there should be an ablation regarding potential biases when using language models.

**Summary Of The Paper:**

The authors create a novel approach for symbolic regression using two different mechanisms. The first is a math language model, and the second is an adaptable strategy for alternating fitness functions during evolution. They experiment with their approach in synthetic and real-world data and achieve state-of-the-art results.

**Summary Of The Review:**

- The work presented here is interesting and novel. There is margin for minor and major improvements in this work. Specially regarding the reproducibility of their training procedure and some more intuition on selecting some parts of the main contribution.

---

> ### Author Response · Authors · 2022-11-19
> **Response to reviewer jSK3's feedback**
>
> We thank reviewer jSK3 for the valuable feedback and recommendations. We address the main concerns below:
>
> 1.  “(major) The authors motivate their work on the premise that deep learning methods are black-box models that remove the possibility of insights when looking into SR. However, later in the introduction the authors admit that using deep learning models is quite good for generalization propose. Their method also uses an RNN as the backbone, which to me seems that would introduce the problem of black-box models into SR again.”
>
> We thank the reviewer for the interesting point. While the process of deriving the equation is a “black-box” process, the equation itself is not. Even in traditional GP-SR, the GP process is still a “black-box”, but the eventual equation that is found is not.
>
> 2. “(minor) Since we are using language models, I wish there were details on why not using Transformers/BERT or similar.”
>
> We thank the reviewer for the suggestion. We have now included results of a transformer model in Table 2 as part of our ablation study. We selected the RNN architecture due to a slight outperformance over the transformer. Nonetheless, the increase of the performance of the transformer over vanilla GP demonstrates that the effectiveness of TW-MLM is not overly reliant on the architecture, as shown by the results.
>
> 3. “I wish the authors had added the methods with human heuristics in their comparison. It would be ideal to have an ablation study on the trade-off from creating heuristics manually (and perhaps adding bias) and machine-made.”
>
> We thank the reviewer for the suggestion, we will highlight this in our revision, in paragraph 1 of section 3.2. For DSR, performance drops sharply with the removal of in-situ constraints and complexity scores, with top-1 approximate recovery rate decreasing to 33% the original value and top-1 exact recovery rate decreasing to 10% the original value.
>
> 4. “The authors are clear about their method and indeed present a novel approach. However, the lack of information regarding the Language Model training procedure hurts reproducibility.”
>
> We thank the reviewer for the comments. We have updated the method description to be clearer and provided the architecture of the layers in the neural network. We have also provided implementation code with settings in the revision.

---

> > ### Comment · Reviewer_jSK3 · 2022-11-29
> > **Response to authors**
> >
> > Thank you for the clarifications, making me more confident on my final recommendation.

---

### Official Review · Reviewer_vhnA · 2022-11-02

**Confidence:** 3
**Correctness:** 2
**Technical Novelty And Significance:** 3
**Empirical Novelty And Significance:** 2
**Recommendation:** 3

**Clarity, Quality, Novelty And Reproducibility:**

* The paper is not well organized. The reviewer feels that it is hard to understand the main focus of this paper and the detailed algorithm of the proposed method.
* The experimental evaluation is not convincing. The ablation study should be done, and the hyperparameter sensitivity of the proposed method should be checked.
* The code is not provided, and it is hard to understand the detailed algorithm of the proposed method.


**Strength And Weaknesses:**

**Strengths**
* Two improving methods for symbolic regression are introduced. The experimental results show that the proposed method is superior to other SR methods, especially on the functions with unknown constants.

**Weaknesses**
* As only the concept-level explanation of the proposed method is given, it is hard to understand and re-implement the detailed algorithm. For instance, the training procedure of RNN and hyperparameter settings of the proposed method are omitted.
* This paper presents two improving methods: pre-trained TW-MLM and MSEDI. However, it is not clear the contribution of each method to the performance gain. The ablation study should be conducted.
* The proposed method uses the pre-trained model on physics equations. A simple baseline method using the physics equation dataset should be considered, e.g., incorporating the same physics equations into the initial population in genetic programming.
* MSEDI assumes a scalar input variable. It seems hard to apply the proposed method to multi-dimensional input problems.

**Comments**
* Why didn't the authors use all datasets in SRBench? The selection reason for the 6 datasets should be clarified.
* The physics equations in Table 7 include various different input variables, while the target functions in the experiments are composed of a single variable. The reviewer could not understand how to handle such a variable mismatch.

**Summary Of The Paper:**

This paper presents two improving methods for symbolic regression (SR). The first method introduces the pre-trained recurrent neural network (RNN) model using the physics equation library to generate the candidate solutions. The second method is a novel loss function called mean squared error of difference (MSEDI). The MSEDI is alternately used with normal MSE in the proposed method. The experimental evaluation using benchmark datasets demonstrates that the proposed method outperformed existing SR methods in several problems.

**Summary Of The Review:**

Although the concept of the proposed approach seems to be interesting, the experimental evaluation is weak to validate its effectiveness. Moreover, the algorithm description should be improved.

---

> ### Author Response · Authors · 2022-11-19
> **Response to reviewer vhnA's feedback**
>
> We thank reviewer vhnA for the valuable feedback and recommendations. We address the main concerns below:
>
> 1. “The paper is not well organized. The reviewer feels that it is hard to understand the main focus of this paper and the detailed algorithm of the proposed method.” And “As only the concept-level explanation of the proposed method is given, it is hard to understand and re-implement the detailed algorithm.”
>
> Thank you for the suggestions. We have updated the paper to be more clear about the objectives and contributions of the paper. Specifically, we foreshadow our proof-of-concept in greater detail starting right from the introduction, added an algorithm flowchart and provided clarifications to the proposed method. Additionally, we have now highlighted the subsections which include the ablation studies and also provided code with settings in the revision. We hope the revised paper is more readable.
>
> 2. “This paper presents two improving methods: pre-trained TW-MLM and MSEDI. However, it is not clear the contribution of each method to the performance gain. The ablation study should be conducted.”
>
> Thank you for the suggestions. We did conduct some experiments to study the contribution of each method. Table 2 shows the improvements from using TW-MLM and the last paragraph of Section 3.3 demonstrates the effectiveness of MSEDI. However, we agree that clearer ablation results are required. As such, we have revised the paper to highlight the subsections that present ablation results (as suggested by reviewers vhnA and jSK3).
>
> 3. “The proposed method uses the pre-trained model on physics equations. A simple baseline method using the physics equation dataset should be considered, e.g., incorporating the same physics equations into the initial population in genetic programming”
>
> We thank the reviewer for the suggestion. Through our new evaluation, this baseline method performs poorly, with high variance in the performance. For example, for the dataset Nguyen-4, it was unable to recover the equation within 100 trials, while all the other methods could. On the other hand, it managed to recover Nguyen-8 in 94% of the experiments. This suggested method is heavily biased to the equation dataset and is poor to generalize to new forms of equations.
>
> 4. “MSEDI assumes a scalar input variable. It seems hard to apply the proposed method to multi-dimensional input problems.”
>
> We thank the reviewer for the question and we have revised the paper to clarify this. The variables in the equations in Table 7 are all represented as a single “variable token” (i.e., there is no differentiation between variable w and variable x when training the TW-MLM). This is done because we wanted TW-MLM to be generalizable for problems with different numbers of variables. When TW-MLM generates a new equation, the “variable token” is replaced by a variable uniformly selected from the variables found in the numerical dataset.
>
> 5. “Why didn't the authors use all datasets in SRBench? The selection reason for the 6 datasets should be clarified.”
>
> We thank the reviewer for the question. We have revised the paper and added in our rationale for doing so. The main issue is the large time complexity with respect to the number of variables. For example, it takes 12400 computer hours to evaluate DSO-NGGP across all the 122 datasets in SRBench, with only 10 experiments conducted per dataset. Likewise, GPLearn takes 8200 computer hours for the same amount of experiments. We selected 6 low-variable datasets and ran 100 experiments per dataset to provide more statistically valid results per dataset.

---

### Author Response · Authors · 2022-11-19
**Changes made to paper to address reviewer feedback**

We thank the reviewers for their feedback and suggestions. We provide a high-level summary of the changes that we've made to the draft to address your feedback:
1. We have reorganized the paper for greater clarity. We foreshadow our proof-of-concept in greater detail starting right from the introduction, add an overall algorithm flowchart and provide clarifications requested by the reviewers.
2. We have taken advice from reviewers to update the method description to be clearer and provide more discussion for the results section, while reducing the contents for the reflections.
3. We have conducted an ablation study to evaluate the performance of TW-MLM with a transformer architecture instead of RNN architecture. We present these results in the revised paper and highlight our existing ablation studies (with and without TW-MLM and with and without the alternating fitness functions).
4. We have also provided code in the supplementary materials for this round.

Specific reviewer comments and questions are addressed below (with new results when applicable).

---

### Decision · Program_Chairs · 2023-01-20

**Decision:**

Accept: poster

**Justification For Why Not Higher Score:**

The problem is not a huge research topic, so might have a limited audience. On the other hand, it is a good paper.

**Justification For Why Not Lower Score:**

Clearly novel, good results, well written.

**Metareview: Summary, Strengths And Weaknesses:**

This interesting paper presents a new approach to symbolic regression (genetic programming to find expressions that conform to datasets). The two innovations is using a language model instead of heuristics to get equations of the right shape (biasing the search space towards equations that look like known equations) and alternating fitness functions between generations to increase adaptability. I find both of these ideas intriguing (with the language model idea probably being the more novel one) and it is satisfying to see that these modifications work and bring strong performance. I think this paper is a clear accept.

**Note From Pc:**

if the above contains the word "oral" or "spotlight" please see: "oral" presentation means -> notable-top-5% and "spotlight" means -> notable-top-25%. As stated in our emails, we are disassociating presentation type from AC recommendations